# The Association between Race/Ethnicity and Cancer Stage at Diagnosis of Bone Malignancies: A Retrospective Cohort Study

**DOI:** 10.3390/ijerph192315802

**Published:** 2022-11-28

**Authors:** Ayman Oweisi, Moawiah S. Mustafa, Luai S. Mustafa, Alyssa N. Eily, Pura Rodriguez de la Vega, Grettel Castro, Noël C. Barengo

**Affiliations:** 1Department of Translational Medicine, Herbert Wertheim College of Medicine, Florida International University, Miami, FL 33199, USA; 2Department of Global Health, Robert Stempel College of Public Health and Social Work, Florida International University, Miami, FL 33199, USA; 3Faculty of Medicine, Riga Stradiņš University, LV-1007 Riga, Latvia

**Keywords:** bone malignancy, cancer stage, disparity, race, ethnicity

## Abstract

Introduction and objective: Limited data exists analyzing disparities in diagnosis regarding primary bone neoplasms (PBN). The objective of our study was to determine if there is an association between race/ethnicity and advanced stage of diagnosis of PBN. Methods: This population-based retrospective cohort study included patient demographic and health information extracted from the National Cancer Institute Surveillance, Epidemiology, and End Results Program (SEER). The main exposure variable was race/ethnicity categorized as non-Hispanic white (NH-W), non-Hispanic black (NH-B), non-Hispanic Asian Pacific Islander (NH-API), and Hispanic. The main outcome variable was advanced stage at diagnosis. Age, sex, tumor grade, type of bone cancer, decade, and geographic location were co-variates. Unadjusted and adjusted logistic regression analyses were conducted calculating odds ratios (OR) and corresponding 95% confidence intervals. Results: Race/ethnicity was not statistically significantly associated with advanced-stage disease. Adjusted OR for NH-B was 0.94 (95% CI: 0.78–1.38), for NH-API 1.07 (95% CI: 0.86–1.33) and for Hispanic 1.03 (95% CI: 0.85–1.25). Conclusions: The lack of association between race and advanced stage of disease could be due to high availability and low cost for initial management of bone malignancies though plain radiographs. Future studies may include socioeconomic status and insurance coverage as covariates in the analysis.

## 1. Introduction

The term bone cancer encompasses several neoplastic malignancies that can affect both the axial and peripheral skeleton. Bone malignancies can also be subdivided into osteosarcomas, Ewing sarcoma, chondrosarcoma, and other unspecified forms [1,2,3]. According to the SEER database, the average age of diagnosis for bone and joint cancers in the United States from 2010–2014 was 43 years of age, owing to a highly bimodal distribution [3]. Approximately 3260 cases of bone cancer are diagnosed annually and the 5-year survival rate of bone cancers from 2007–2013 in the United States was about 67.7% [3]. Detection of several molecular and cytogenetic aberrations have resulted in new therapeutic targets for bone malignancies; however, the 5-year survival rate has remained stagnant since the 1990s [4,5]. Given this stagnation, epidemiological factors must be considered as critical variables in prognosis.

The relationship between ethnic disparities in prognoses and clinical outcomes in breast, colorectal, and gynecological malignancies has been previously established [6,7,8,9,10,11,12]. Diagnosing cancer at an equivalent stage, regardless of race, may lead to similar survival rates if other confounding factors, such as socioeconomic status and payer status, are controlled [10]. Racial bias may play a significant role in a physician’s likelihood to treat a patient or investigate symptoms of potential malignancy as seen in the treatment of pediatric asthma, pain management, and thrombolysis [8,9]. A later stage of diagnosis for cancer among black women when compared to white women has also been shown to cause a decrease in compliance and acceptance of offered treatment options [8]. Furthermore, the stage of diagnosis for malignancies is the most important aspect of the prognosis and therefore, crucial to the patient outcomes [13].

Previous studies have shown an association between race/ethnicity and stage of diagnosis of PBN. However, no studies have completed a combined analysis using the most common types of PBN [1,2,14,15,16]. Given the rarity of PBN, studies focusing on singular bone neoplasms may lack sufficient sample size and power to assess for true differences. Additionally, several studies have evaluated differences in overall survival without analyzing factors, such as stage at diagnosis, which may contribute to racial disparities. In addition, we are the first study to include the differences in stage of diagnosis of PBN of individuals of different race/ethnicity from various decades as well as geographic location.

The objective of our study was to determine if there is an association between race/ethnicity and advanced stage of diagnosis of PBN.

## 2. Materials and Methods

### 2.1. Study Design and Population

This population-based retrospective cohort used the National Cancer Institute’s Surveillance, Epidemiology, and End Results Program (SEER). SEER is a cancer surveillance program of the National Cancer Institute’ s (NCI) that collects cancer incidence and survival data since 1973, including data from the Atlanta, Detroit, San Francisco-Oakland, and Seattle-Puget Sound metropolitan areas, and the states of Connecticut, Hawaii, Iowa, New Mexico, and Utah [17].

The study population consisted of adult participants of the SEER Program aged 18 to 85 years with a new diagnosis and confirmed cases of primary bone neoplasm (osteosarcoma, Ewing sarcoma, chondrosarcoma, and giant cell tumor) from 1973–2014. Subject data was divided into cohorts of approximately one decade: 1973–1982, 1983–1992, 1993–2002, 2003–2014. Patients were excluded under the following conditions: cancer diagnosed at autopsy, race classified as “American Indian/Alaskan Native” or “unknown”; and unstaged cancer patients.

### 2.2. Study Variables

The outcome was clinical cancer staging at diagnosis, with the categorization based on SEER summary staging. Outcomes were dichotomized as either early stage (localized only) or late stage (regional spread by direct extension only, and regional spread by both direct extension and lymph node involvement) at diagnosis.

The main exposure variable was race/ethnicity defined as Non-Hispanic (NH) Whites, NH-Bs, NH-Asian/Pacific Islander (API) and Hispanic, created based on the variables SEER race recorded and origin recorded (Hispanic, Non-Hispanic).

There are several potential confounders with could be associated with our study, including age, sex, year of diagnosis, histological types and ICD-O-3 codes, tumor grade, and geographical location. The age was reported as a continuous variable and divided into three discrete categories (pediatric: ≤21; adult: 22–65; geriatric: 65–85). This could present as a confounder in our study due to various PBNs often occurring in children and young adults between ages 10–40 years old. Similarly, studies have shown that sex plays a role in the development of various bone malignancies, with males being slightly more at risk for development of a PBN. Geographic location plays a significant role in the progression of PBNs due to individuals in rural communities typically having less access to healthcare and preventative screening as opposed to urban areas.

### 2.3. Statistical Analysis

Stata version 14 (StataCorp, College Station, TX, USA) was used for the statistical analysis. First, descriptive statistics were applied examining frequency distributions of the categorical variables and assessing for missing data. Then, Chi-square tests were used to compare the distribution of the baseline characteristics (demographics and clinical variables) according to race/ethnicity. Collinearity testing was conducted to examine whether two or more variables were highly correlated with each-other. Finally, unadjusted and adjusted odds ratios (OR) and their 95% confidence intervals (CI) were calculated using binary logistic regression. Statistical significance was considered for *p* value < 0.05.

### 2.4. Ethical Considerations

All data accessed from SEER was de-identified (fully anonymized) and without any of the 18 direct identifiers according to the Health Insurance Portability and Accountability Act. Thus, IRB approval was waived by the Florida International University.

## 3. Results

Table 1 shows the distribution of specific covariates across localized and regional/metastatic disease. The results indicate no association between race/ethnicity and stage at diagnosis (*p* = 0.536). Adults aged 22–64 had the lowest proportion of metastatic disease at 51.1%. Stage at diagnosis also varied between sexes (*p* < 0.001) with 61.1% of males presenting with advanced stage disease compared to 52.8% of females. Additionally, stage at diagnosis varied with tumor grade (*p* < 0.001). Grade III–IV tumors had higher proportions of patients with advanced-stage disease (65.2% and 69.4% respectively) compared to grades I–II (29.9%, 47.8%). A significant proportion of advanced-stage tumors were classified as not-determined (63%). The percentage of regional/distant disease was different between types of PBN (*p* < 0.001). Both osteosarcoma and Ewing sarcoma had considerably higher frequencies of patients with metastatic disease (64.9% and 68.2%, respectively) than chondrosarcoma or giant cell tumor (42.1% and 48.2%, respectively). Finally, stage at diagnosis did not vary with decade of diagnosis or geographic location (*p* = 0.15, respectively *p* = 0.425).

Table 2 shows the unadjusted and adjusted odds ratio between race/ethnicity and advanced-stage disease. In both the unadjusted and adjusted analysis, race/ethnicity was shown to have no association with advanced-stage disease. When adjusting for covariates, the odds of being diagnosed with advanced-stage disease increases by 66% (95% CI: 1.42–1.96) in the geriatric population when compared to patients 22–64 years-of-age. Additionally, male sex was associated with a 1.35-fold increase in odds of presenting with advanced-stage disease compared to females (OR = 1.35, 95% CI: 1.22–1.51). All tumors grade II and above were more likely to have regional/distant metastases at the time of diagnosis. Specifically, grade IV tumors had a 370% increase of advanced-stage diagnosis compared to grade 1 (OR = 3.7, 95% CI: 2.92–4.69). Ewing sarcoma was associated with 28% increased odds of presenting with metastases when compared to osteosarcoma after adjusting for confounders (OR = 1.28, 95% CI: 1.10–1.49). There was no significant change in odds ratio across time or geographic areas in the adjusted analysis.

## 4. Discussion

Our study did not show an association between race/ethnicity and advanced-stage disease. The most important secondary finding revealed that men were much more likely to present with advanced-stage disease compared to women.

Our results regarding the association between race/ethnicity and stage of diagnosis of PBN are discordant with previous literature that demonstrated significant differences in outcomes among the different race/ethnicities [1,2,14,15,16]. Previous studies have shown disparities between race and severity of disease presentation in sarcomas. Two studies showed worse overall survival of non-Hispanic black patients with bone sarcomas when compared to white and Hispanic cohorts [14,16]. According to Worch et al. [16], black patients with Ewing sarcoma showed an 11.5% decrease in overall survival (OS) compared to that of their white counterparts (40.7% vs. 52.2%, *p* = 0.015). A similar analysis of the SEER database by Joseph et al. (2017) revealed a significant difference in stage of disease and race/ethnicity [14]. In a cross sectional, population-based retrospective cohort, analysis of 1261 cases of patients 20 years and younger with soft tissue sarcomas, univariate analysis showed non-Hispanic blacks have lower OS compared to non-Hispanic whites (HR = 1.71, 95% CI: 1.25–2.34) [14]. While this association was mitigated in a multivariate analysis, non-Hispanic blacks were 63% more likely to present with advanced stage disease with distant metastases being the most significant predictor of OS (HR = 6.12, 95% CI: 4.48–8.40).

Jacobs et al. showed OS was better for white patients than black or Hispanic when considering both soft tissue and bone sarcomas. The disparity became most pronounced in the 1993–2002 (white (72.9%) vs Hispanic (68.6%) or black patients (67.5%), *p* < 0.001) [1]. Analysis showed a trend in disparities between white and black or Hispanic patient 5-year survival in bone sarcomas alone; however, this did not reach statistical significance (*p*-value not provided) [1]. The inferior OS of non-Hispanic black patients persists when controlling for demographic and disease-related factors. Two studies demonstrated higher age-adjusted incidence rates of osteosarcoma or Ewing sarcoma in non-Hispanic black patients [2,15]. According to two studies, non-Hispanic black patients are more likely to present with advanced-stage disease [1,14]. Duong et al. noted a bimodal distribution of appendicular osteosarcoma with peaks between 10–19 years old and greater than 80 years old [2]. Axial malignant osteosarcomas included an additional incidence peak between ages 40–49 [2]. Karski et al. did not evaluate frequency but did demonstrate that despite only 13.8% of patients with Ewing sarcoma being ≥ 40, those patients are more likely to present with metastases at diagnosis [15].

Combining the data for all PBNs may have underestimated the possible relationship between race and advanced stage of cancer in the former two diseases. It is important to consider the diagnostic protocol for primary bone malignancies. CT, MRI, and PET imaging are classically utilized to investigate the character and extent of a PBN. However, the initial imaging utilized during the workup of a possible PBN is a plain radiography [5]. PBN invariably present with a painful and poorly marginal lesion on plain radiograph, which urges the clinician to pursue more sophisticated diagnostic measures [5]. X-rays have become a widely available and relatively inexpensive means to screen patients for PBN, which likely played a role in minimizing possible racial disparities involved in diagnosing PBNs. In addition, the rich sensory innervation of bones that lends to the primary symptom of pain in PBNs also invites clinicians to initiate the workup with a plain radiograph at an early stage in the initial evaluation. The results of this study support such inferences by demonstrating that the stage at diagnosis did not vary with decade of diagnosis or geographic location. The fact that 5-year survival rates have remained stagnant since the 1990s for bone malignancies could also indicate that methods of early detection have not significantly improved [3,6]. Beyond racial disparities, there may be a paramount underlying association with a later stage of diagnosis, such as average annual income or insurance status. Unfortunately, such information has only recently been included in the records of the SEER database and was insufficiently reported to include in this study.

To date, there is limited research on proposed mechanisms between race/ethnicity and advanced-stage diagnosis of PBN. Mutations in the retinoblastoma protein (pRB) gene are associated with both retinoblastoma and osteosarcoma. Although the SEER database does not include information on confirmed mutations, multifocal unilateral or bilateral disease can be used as a surrogate for genetic testing [7]. A previous analysis of the cases of heritable retinoblastoma in the SEER database demonstrated higher incidence of RB in males than females and non-Hispanic black compared to non-Hispanic whites [7]. Additionally, it has been hypothesized that pRB dysregulation impacts integrin and cadherin-mediated adhesion resulting in increased rates of metastasis [8]. Additionally, low socioeconomic status and insurance are associated with increased time to seeking medical care [9,10]. Non-Hispanic blacks and Hispanics are more likely to be underinsured than non-Hispanic whites and to live in areas with poor healthcare access [11]. It is possible that advanced disease stage in minorities may be associated with decreased access to medical care.

One of the main limitations of our study relates to the point mentioned above. We were unable to account for socioeconomic status (SES) and insurance status. Unfortunately, both SES and insurance status are inextricably related to race/ethnicity across America [12,13]. As a result, our study omitted potential confounders that may have impacted the odds ratio. Another confounder that may influence the study outcome would be physical exercise level in the participants. Physical activity has been shown to reduce levels of pro-inflammatory biomarkers, and there have been studies that reveal lower levels of CRP in physical active breast cancer survivors [18]. Individuals that have a higher activity level have demonstrated that there is an improvement of quality of life, even when being treated for a PBN, with improvements seen in muscle strength, pain, fatigue, anxiety, depression, and loss of joint mobility. Physical exercise has shown to not only improve symptoms of malignancy and adverse medication effects, but it has also shown to reduce risk factors and mortality in this patient population [19]. In addition, another limitation of this study involves the reliance on a secondary data set with a large amount of missing data points, particularly prior to 2004. However, the SEERs database has a large sample size and length of follow-up necessary to investigate rare diseases, such as PBN. However, multiple variables beyond the advanced stage that impacted overall survival were not included in our statistical analysis. Non-Hispanic black patients had significantly higher proportions of high-grade disease which, in the context of the association between high grade and odds of metastases at diagnosis, indicates that non-Hispanic black patients are likely at increased risk of presenting with advanced stage disease. We were unable to find previous studies that combined multiple bone neoplasms in their analysis. Given the skewed distribution of PBN with a large majority classified as osteosarcoma, future statistical analysis should stratify patients based on specific PBN.

## 5. Conclusions

In conclusion, the increased proportion of PBN across time may be worrisome for potential increased genetic and environmental factors contributing to bone malignancy. However, we may also consider that this increase can be better explained by improvement in diagnostic techniques including imaging and cytometry. Geriatric patients had significantly higher odds of presenting with advanced-stage disease, which is significant considering previous literature demonstrating that non-Hispanic blacks have a higher age-adjusted incidence of osteosarcoma [1]. Extrapolating from this data, the prognosis of geriatric patients with osteosarcomas is quite poor when compared with younger patients. Both unadjusted and adjusted analysis showed patients with Ewing sarcoma are more likely to present with advanced-stage disease. The clinical significance of this finding cannot be overstated considering the 5-year survival rate of ES drops precipitously from greater than 70% to less than 30% when comparing localized to metastatic disease. Physicians should maintain a high index of clinical suspicion for PBN in patients with consistent, progressive pain and/or rapidly growing masses in all patients but specifically in minority patients whose pain is often minimized by providers. Maintaining a high index of clinical suspicion can potentially reduce advanced-stage diagnosis of PBN across all ages and result in improved survival. One also considers quality of life irrespective of overall survival with regards to potential radiation and expansive amputations associated with advanced-stage disease. Future studies should control their analysis for socio-economic factors as well when assessing the association between race and cancer stage at diagnosis.

## Figures and Tables

**Table 1 ijerph-19-15802-t001:** Distribution of race/ethnicity and potential confounders by stage of diagnosis in the United States from 1973–2014.

		Stage			
Characteristics	Localized (*n* = 2667)		Regional/Distant (*n* = 3597)		*p*-Value
	*n*	%	*n*	%	
Race/Ethnicity					0.536
NH-White	2053	43.0	2723	57.0	
NH-Black	230	42.2	315	57.8	
NH-API	156	39.4	240	60.6	
Hispanic	228	41.7	319	58.3	
Age (years)					<0.001
0–21	902	36.8	1547	63.2	
22–64	1419	48.9	1483	51.1	
65–85	346	37.9	567	62.1	
Sex					<0.001
Male	1368	38.9	2145	61.1	
Female	1299	47.2	1452	52.8	
Tumor Grade					<0.001
Grade I	619	71.1	252	29.9	
Grade II	468	52.2	528	47.8	
Grade III	236	34.8	442	65.2	
Grade IV	313	30.6	711	69.4	
Not-determined	1031	36.9	1764	63.1	
Type of Bone Cancer					<0.001
Osteosarcoma	915	35.1	1689	64.9	
Chondrosarcoma	1245	57.9	906	42.1	
Giant Cell Tumor	70	51.9	65	48.2	
Ewing Sarcoma	437	31.8	937	68.2	
Decade					0.15
1973–1982	504	43.1	666	56.9	
1983–1992	511	39.9	770	60.1	
1993–2002	720	44.0	916	56.0	
2003–2014	932	42.8	1245	57.2	
Geographic Location					0.425
Completely/Mostly Rural	161	40.7	235	59.3	
Mostly Urban	2506	42.7	3362	57.4	

**Table 2 ijerph-19-15802-t002:** Unadjusted and adjusted odds ratio between characteristics and stage at diagnosis in United States from 1973–2014.

Characteristics	Unadjusted	Adjusted ^1^
	OR ^2^ (95% CI ^3^)	OR (95% CI)
Race/Ethnicity		
NH-W (*n* = 4776)	Ref ^4^	
NH-B (*n* = 545)	1.03 (0.86–1.24)	0.94 (0.78–1.38)
NH-API (*n* = 396)	1.16 (0.94–1.43)	1.07 (0.86–1.33)
Hispanic (*n* = 547)	1.05 (0.88–1.26)	1.03 (0.85–1.25)
Age (years)		
0–21 (*n* = 2449)	1.64 (1.47–1.83)	0.90 (0.79–1.03)
22–64 (*n* = 2902)	Ref	
65–85 (*n* = 913)	1.57 (1.35–1.83)	1.66 (1.42–1.96)
Sex		
Female (*n* = 2070)	Ref	
Male (*n* = 2706)	1.40 (1.27–1.55)	1.35 (1.22–1.51)
Tumor Grade		
Grade I (*n* = 714)	Ref	
Grade II (*n* = 716)	2.25 (1.85–2.73)	2.14 (1.75–2.60)
Grade III (n = 489)	4.60 (3.71–5.71)	3.32 (2.61–4.21)
Grade IV (*n* = 690)	5.58 (4.58–6.80)	3.70 (2.92–4.69)
Not determined (*n* = 2167)	4.20 (3.56–4.96)	2.73 (2.21–3.36)
Type of Bone Cancer		
Osteosarcoma (*n* = 1766)	Ref	
Chondrosarcoma (*n* = 1784)	0.39 (0.35–0.44)	0.57 (0.48–0.68)
Giant Cell Tumor (*n* = 89)	0.50 (0.36–0.71)	0.54 (0.38–0.78)
Ewing Sarcoma (*n* = 1137)	1.16 (1.01–1.34)	1.28 (1.10–1.49)
Decade		
1973–1982 (*n* = 977)	0.99 (0.86–1.14)	0.99 (0.85–1.17)
1983–1992 (*n* = 1017)	1.13 (0.98–1.30)	1.15 (0.99–1.34)
1993–2002 (*n* = 714)	0.95 (0.84–1.08)	0.99 (0.87–1.14)
2003–2014 (*n* = 714)	Ref	
Geographic Location		
Mostly urban (*n* = 4403)	Ref	
Completely/Mostly Rural (*n* = 373)	1.09 (0.88–1.34)	

^1^ Variables included in the adjusted model were race/ethnicity, age, sex, tumor grade, type of bone cancer, and decade. ^2^ Odds ratio. ^3^ Confidence interval. ^4^ Reference standard to which other groups were compared.

## Data Availability

National Cancer Institute: Surveillance, Epidemiology, and End Results (SEER) Available: http://seer.cancer.gov, accessed on 18 October 2018.

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
