# Peer review of "The Association between Race/Ethnicity and Cancer Stage at Diagnosis of Bone Malignancies: A Retrospective Cohort Study"

_ijerph, 2022, doi:10.3390/ijerph192315802_

Round 1

Reviewer 1 Report

In this paper, authors tried to determine whether there is an association between race and advanced stage of primary bone neoplasms (PBN) by analyzing the dataset from NCI. Although the final results indicated that there is not a statistically significant association between race and stage at diagnosis, I think this paper also somewhat provides a guideline for clinic therapy.  

In my opinion, this paper may be accepted for publication after the authors address the questions below. 

Major 

  1. In the end, authors suggested that “Future studies should consider a regression model to assess the impact of race/ethnicity, SES, age, sex and type of cancer on stage at diagnosis”. That is a very good point. I think the present work of this study is too light to be published. So authors should also at least analyze the association of age and advanced stage of PBN, and the association of sex and advanced stage of PBN. 

  1. Although previous data had shown the relationship between overall survival and race in Ewing sarcoma patients and chest wall sarcoma patients. they are just partd of bone malignancies. I think in this paper, it is necessary to analyze the association of survival and race in patients among all kinds of bone cancers. 

Minor 

  1. Line 56, 148, 177, 180, 181, 182, 185, 186, 187, etc, I think the word “PBM” should be replaced by “PBN”

Author Response

Major Comments

Comment #1: In the end, authors suggested that “Future studies should consider a regression model to assess the impact of race/ethnicity, SES, age, sex and type of cancer on stage at diagnosis”. That is a very good point. I think the present work of this study is too light to be published. So authors should also at least analyze the association of age and advanced stage of PBN, and the association of sex and advanced stage of PBN.

Response #1: Thank you for your comment. We think there may be a misunderstanding. Unadjusted and adjusted logistic regression models were used for the analysis (Table 2). The adjusted model already included age, sex, tumor grade, type of bone cancer, decade, and geographical region. What we wanted to say is that future studies should additionally control for socio-economic factors as this information was not available to us in our study. We have revised the sentence in the conclusion section that states now as follows:

“Future studies should control their analysis for socio-economic factors as well when assessing the association between race and cancer stage at diagnosis. “

Comment #2: Although previous data had shown the relationship between overall survival and race in Ewing sarcoma patients and chest wall sarcoma patients. They are just parts of bone malignancies. I think in this paper, it is necessary to analyze the association of survival and race in patients among all kinds of bone cancers.

Response #2: We would like to clarify that our study included all bone malignancies (osteosarcoma, Ewing sarcoma, chondrosarcoma, and giant cell tumor) from 1973-2014. These bone cancers were specifically chosen due to these being common types of primary bone neoplasms. Other bone malignancies such as chondrosarcomas are much rarer and do not have sufficient study data through NCI SEER.

Minor Comments

Comment #3: Line 56, 148, 177, 180, 181, 182, 185, 186, 187, etc, I think the word “PBM” should be replaced by “PBN”

Response #3: Thank you very much. We have replaced PBM by PBN accordingly.

Author Response

Major comments

Title

Comment #1: The objective was to investigate the association between race/ethnicity and advanced-stage bone malignancies. However, the title (The Association Between Race/Ethnicity and Cancer Stage at Diagnosis of Bone Malignancies) was somewhat misleading. The readers may assume that the manuscript reports the association between race/ethnicity and different stages of bone malignancies. Moreover, the paper reports the associations between several factors (race/ethnicity, age, sex, tumor grade, type of bone cancer, decade, and geographic region), not only race/ethnicity, and advanced stage at diagnosis.

Response #1:

I suggest:

“The associations between socio-demographic characteristics and cancer stage at diagnosis of bone malignancies: A retrospective study”

Comment #2: Please indicate the study design in the title (i.e., The Association Between Race/Ethnicity and Advanced-Stage Bone Malignancies: A Retrospective Cohort)

Response #2: Title was adjusted to reflect the study design. The title reads now as follows:

“The Association Between Race/Ethnicity and Cancer Stage at Diagnosis of Bone Malignancies: A Retrospective Cohort Study”

Introduction

Comment #3: It would be clearer if the authors could explicitly mention the difference between the present study and previous studies on the topic, what could not be addressed by the previous studies but could be addressed by the present study, and the necessity, novelty, and potential contribution of the present study.

Response #3:

We have revised the third paragraph of the introduction that reads now as follows:

“Previous studies have shown an association between race/ethnicity and stage of diagnosis of PBN. However, no studies have included the most common types of PBN in their analysis [1-2, 14-16]. Given the rarity of PBN, studies focusing on singular bone neoplasms may lack sufficient sample size and power to assess for true differences. Additionally, several studies have evaluated differences in overall survival without analyzing factors such as stage at diagnosis that may contribute to racial disparities. In addition, we are the first study to include the differences in stage of diagnosis of PBN of individuals of different race/ethnicity from various decades as well as geographic location. ”

Methods

Study design and population

Comment #4: Why were American Indians/Alaskan Natives excluded?

Response #4: These patients were excluded from the study due to small sample size on SEER NCI as well as no statistics for women in this category.

Study variables

Comment #5: The outcome was not clearly defined. How to define late/advanced-stage disease?

Response #5: Advanced stage PBN was defined in the study variables as having either regional spread by direct extension or regional spread through both direct extension or lymph node involvement. We have defined this now in the methods section. The following sentence was added:

“Outcomes were dichotomized as either early stage (1: Localized only) or late stage (2: Regional spread by direct extension only, and 4: Regional spread by both direct extension and lymph node involvement) at diagnosis.”

Comment #6: Please clearly define potential confounders and effect modifiers.

Response #6:  We have revised the paragraph that describes the potential confounders/covariates included in this study. The new paragraph reads now as follows:

Covariates included:

“There are several potential confounders which could be associated with our study, including age, sex, year of diagnosis, histological types and ICD-O-3 codes, tumor grade, and geographical location. The age was reported as a continuous variable and divided into three discrete categories (pediatric </=21; adult >/=22, <65; geriatric >/=65, </=85). This could present as a confounder in our study due to various PBNs often occurring in children and young adults between ages 10-40 years old. Similarly, studies have shown that sex plays a role in the development of various bone malignancies, with males being slightly more at risk for development of a PBN. Geographic location plays a significant role in the progression of PBNs due to individuals in rural communities typically having less access to healthcare and preventative screening as opposed to those living in urban areas. “

Comment #7: “Age reported as a continuous variable and divided into three discrete categories (pediatric </=21; adult >/=22, <65; geriatric >/=65, </=85)”. If age 18 is considered an adult. How age ≤ 21 was considered pediatric?

Response #7: We agree with the reviewer and have corrected the categories as follow:

“Age reported as a continuous variable and divided into three discrete categories (</=21;>/=22, <65; >/=65, </=85)”.

Comment #8: The authors mentioned that diagnosing cancer at an equivalent stage, regardless of race, may lead to similar survival rates if other confounding factors, such as socioeconomic status and payer status, are controlled. Please clarify why the present study did not consider socioeconomic and payer status.

Response #8: We were unable to account for socioeconomic status (SES) and insurance status as that information was not available for us from SEER. We have addressed this in limitations of the study section that reads now as follows:

“We were unable to account for socioeconomic status (SES) and insurance status. Unfortunately, both SES and insurance status are inextricably related to race/ethnicity across America [12-13]. As a result, our study omitted potential confounders that may have impacted the odds ratio. ”

Comment #9: Lifestyle factors could modify the association of interest. Please clarify why lifestyle factors were not considered.

Response #9: Unfortunately, SEER does not collect information on lifestyle habits or risk factors of cancer.

We have added the following information to the limitation section:

“Another confounder that may influence the study outcome would be physical exercise level in the participants. Physical activity has been shown to reduce levels of pro-inflammatory biomarkers and there have been studies shown that reveals lower levels of CRP in physical active breast cancer survivors (18). “

Statistical analysis

Comment #10: There was no description regarding which variables were adjusted in the fully adjusted model.

Response #10: Variables for the adjusted and unadjusted odds ratio were included in the results section as well as table 2. Variables that were adjusted were race/ethnicity, age, sex, tumor grade, type of bone cancer, and decade.

We have added the following information below Table 2:

1Variables included in the adjusted model were race/ethnicity, age, sex, tumor grade, type of bone cancer, and decade.

Comment #11: I think it would be more interesting to investigate the joint association of race/ethnicity and other factors (e.g., age, sex, tumor grade, type of bone cancer, decade, and geographic region) with advanced-stage bone malignancies at diagnosis rather than the association between each of those factors and advanced-stage bone malignancies at diagnosis.

Response #11: Thank you for your suggestion. We hope to address the proposed new and different research question in a future study conducted by our team.

We have added the following sentences to the conclusion section:

“Future studies may also investigate the joint association of race/ethnicity and other factors (e.g., age, sex, tumor grade, type of bone cancer, decade, and geographic region with advanced-stage bone malignancies.”

Results

Comment #12: Non-significant findings should be referred to as no association rather than non-statistically significant.

Response #12: Thank you. We have revised the manuscript according to your suggestion.

Comment #13: Table 2 should include the number of overall participants and cases.

Response #13: We have added that information to Table 2.

Comment #14: Please indicate the variables included in the adjusted model in Table 2.

Response #14: We have added the following information below Table 2:

1Variables included in the adjusted model were race/ethnicity, age, sex, tumor grade, type of bone cancer, and decade.

Discussion

Comment #15: The paper reports the associations between several factors (race/ethnicity, age, sex, tumor grade, type of bone cancer, decade, and geographic region), not only race/ethnicity. However, factors other than race/ethnicity were not mentioned in the abstract, title, introduction, and discussion.

Response #15: We have revised the title. The new title reads as follows:

“The associations between socio-demographic characteristics and cancer stage at diagnosis of bone malignancies: A retrospective study”

Minor comments

Comment #16: Please use either advanced-stage or late-stage consistently throughout the manuscript.

Response #16: We made sure that advanced-stage or late-stage were used consistently throughout the manuscript.

Reviewer 3 Report

Dear Authors,

As reported in literature, ethnicity and social conditions might play a role in orienting the clinicians’ approach to patients. As a consequence, some patients might experiment delays in diagnosis, with burdensome implications both for prognosis and adherence to therapies.

In consideration of the rarity of primitive bone malignancies, there is little evidence concerning the relationship between ethnicity and cancer stage at the time of the diagnosis.

In this scenario, this Manuscript offers an interesting perspective.

However, I think that some issues should be assessed in order to make the present paper suitable for publication.

Major reviews

DISCUSSION: As stated in the Results, ORs presented in this paper were analyzed both unadjusted and adjusted for confounders. However, some confounders were not assessed.

As an example of potential source of confoundment, did you assess physical activity level and its relationship with social factors?

Please discuss this critical issue citing the following references:

-       Invernizzi M, Lippi L, Folli A, Turco A, Zattoni L, Maconi A, de Sire A, Fusco N. Integrating molecular biomarkers in breast cancer rehabilitation. What is the current evidence? A systematic review of randomized controlled trials. Front Mol Biosci. 2022 Sep 8;9:930361. doi: 10.3389/fmolb.2022.930361.

-       Montaño-Rojas LS, Romero-Pérez EM, Medina-Pérez C, Reguera-García MM, de Paz JA. Resistance Training in Breast Cancer Survivors: A Systematic Review of Exercise Programs. Int J Environ Res Public Health. 2020 Sep 7;17(18):6511. doi: 10.3390/ijerph17186511..

-       Garcia MB, Ness KK, Schadler KL. Exercise and Physical Activity in Patients with Osteosarcoma and Survivors. Adv Exp Med Biol. 2020;1257:193-207. doi: 10.1007/978-3-030-43032-0_16.

RESULTS: Apparently, the Results presented in this section have no correspondence with the numbers in Table 1. Please, verify the coherence of data.

Minor reviews

WHOLE MANUSCRIPT: Please, provide reference numbers between square brackets.

INTRODUCTION: Page 2, line 85. Please, correct the typo (“.”).

Author Response

DISCUSSION:

Comment #1: As stated in the Results, ORs presented in this paper were analyzed both unadjusted and adjusted for confounders. However, some confounders were not assessed.

As an example of a potential source of confoundment, did you assess physical activity level and its relationship with social factors?

Please discuss this critical issue citing the following references:

-       Invernizzi M, Lippi L, Folli A, Turco A, Zattoni L, Maconi A, de Sire A, Fusco N. Integrating molecular biomarkers in breast cancer rehabilitation. What is the current evidence? A systematic review of randomized controlled trials. Front Mol Biosci. 2022 Sep 8;9:930361. doi: 10.3389/fmolb.2022.930361.

-       Montaño-Rojas LS, Romero-Pérez EM, Medina-Pérez C, Reguera-García MM, de Paz JA. Resistance Training in Breast Cancer Survivors: A Systematic Review of Exercise Programs. Int J Environ Res Public Health. 2020 Sep 7;17(18):6511. doi: 10.3390/ijerph17186511..

-       Garcia MB, Ness KK, Schadler KL. Exercise and Physical Activity in Patients with Osteosarcoma and Survivors. Adv Exp Med Biol. 2020;1257:193-207. doi: 10.1007/978-3-030-43032-0_16.

Response #1: Included in discussion:

“Another confounder that may influence the study outcome would be physical exercise level in the participants. Physical activity has been shown to reduce levels of pro-inflammatory biomarkers and there have been studies that reveal lower levels of CRP in physical active breast cancer survivors (18). Individuals that have a higher activity level have demonstrated that there is an improvement of quality of life, even when being treated for a PBN, with improvements seen in muscle strength, pain, fatigue, anxiety, depression, and loss of joint mobility. Physical exercise has shown to not only improve symptoms of malignancy and adverse medication effects, but it has also shown to reduce risk factors and mortality in this patient population [19].

RESULTS:

Comment #2: Apparently, the Results presented in this section have no correspondence with the numbers in Table 1. Please, verify the coherence of data.

Response #2: We have no made sure that the text and the table matches.

Table 2. Distribution of race/ethnicity and potential confounders by stage of diagnosis in the United States from 1973-2014.

Minor reviews

WHOLE MANUSCRIPT:

Comment #3: Please, provide reference numbers between square brackets.

Response #3: Manuscript has been adjusted to reflect this change.

INTRODUCTION:

Comment #4: Page 2, line 85. Please, correct the typo (“.”).

Response #4: Thank you. The type has been corrected.

Round 2

Reviewer 2 Report

The authors have adequately addressed most of my comments. I have no further comments.

Reviewer 3 Report

Dear Authors,

in my opinion, the manuscript is interesting, and the results are intriguing.

You have significantly improved the paper during the revision process.

Therefore, in my opinion, the paper is now suitable for publication in this Journal.